

# The baseline wander correction based on improved EEMD algorithm for grounded electrical source airborne transient electromagnetic signals

Yuan Li [1], Song Gao[2,3], Saimin Zhang[1,2], Hu He[3], Pengfei Xian[3], Chunmei Yuan[3]

[1]College of Geophysics, Chengdu University of Technology, Chengdu, 610059, China
[2]Key Laboratory of Earth Exploration and Information Techniques of Ministry of Education, Chengdu, 610059, China
[3]College of Information Science and Technology, Chengdu University of Technology, Chengdu, 610059, China

*Correspondence to*: Yuan Li ( 86210111@qq.com )

**Abstract.** The grounded electrical source airborne transient electromagnetic (GREATEM) system is an important method for obtaining subsurface conductivity distribution as well as outstanding detection efficiency and easy flight control. However, the signals are the superposition of useful signals and various noise signals. The baseline wander caused by the receiving coil motion always exists in the process of data acquisition to affect the measurement results. The baseline wander is one of the main noise sources of data, which has the low frequency, large amplitude, non-periodic and non-stationary and

so on. Consequently, it is important to correction baseline wander for inversion explanation of GREATEM. In this paper, we propose improving method EEMD-AF based on ensemble empirical mode decomposition (EEMD) to correction baseline wander. The EEMD-AF method will decompose the electromagnetic signal into multi-stage intrinsic mode function (IMF) components and adaptively filter high-order IMF component which containing the baseline wander. To examine the performance of our introduced method, we used the EEMD-AF method for the signal baseline correction and compared with

sym8 wavelet with 10 decomposition levels and EEMD with deleted higher-order components directly. The various methods were applied to process the synthetic data and field data. Through the evaluation of the signal-to-noise ratio (SNR) and mean-square-error (MSE), the correction result indicates that the signal using EEMD-AF method can get higher SNR and lower MSE. Comparing correctional signal using the EEMD-AF and the wavelet-based method in the anomaly curves profile images of the response signal, it is proved that the EEMD-AF method is a practical and effective method for removal

of the baseline wander on GREATEM signal.

## 1 Introduction

The grounded electrical source airborne transient electromagnetic (GREATEM) system consists of two parts: the ground transmitter and air receiver. This method takes advantage of Airborne electromagnetic method (AEM) and Magnetotelluric



method (MT), which has large detection depth, higher signal resolution and outstanding detection efficiency (Mogi T,1998;
Smith R S, 2001).

The measured signals are the superposition of useful signals and various noise signals. The noise may be divided into stationary white noise and non-stationary noise which contained sferics noise, human electromagnetic noise and motion-induced noise (Abderrezak et al., 2010; Buselli et al., 1998; Macnae et al., 1984). Sferics noise is mainly caused by the charge discharge in the atmosphere, and the frequency is within 1kHz. Human electromagnetic noise is caused by 50Hz or
60Hz industrial frequencies and its odd harmonics. Motion-induced noise has its own characteristics, such as low frequency, large amplitude, non-periodic and non-stationary. The noise from the receiving coil motion is one of the major noises of the GREATEM signal, which could cause the baseline wander, resulting in signals with a dispersed, non-stationary and low frequency distribution. This phenomenon always exists in the process of data acquisition to affect the measurement results. Severe baseline wanders in the measured EM signal lead to inferior resistivity image formation and affect the reliability of
inversion explanation. After removing sferics noise, human electromagnetic noise and motion-induced noise, processed data will be stacked and averaged on the next stage.

Because the air receiver system is mounted on aircraft such as the rotor-wing unmanned aerial vehicle (UAV) and airship, GREATEM system is different from AEM. First, during the flight, the vibration and speed of the aircraft are weaker than airborne electromagnetic system so that the amplitude of the mounted coil swing is smaller. Hence, the amplitude of
GREATEM caused by baseline wander is smaller than AEM signals. Second, the frequency of the baseline wander signal is narrower than airborne electromagnetic system. The frequency distribution of the motion-induced noise of the AEM is within 1kHz, while the frequency range of GREATEM is mostly within a few Hz in the actual measurement. Third, due to the use of miniaturized airship of GREATEM, the maximum flight loads are much less than the AEM. It is impossible to install the correctional complex mechanical structure on the receiving system to filter out baseline wander.

In the method of filtering out baseline wander, on the one hand mechanical correctional structure and hardware filter can be increased, on the other hand digital filtering and fitting can be used for data processing. Some of studies on the motion-induced noise have focussed on GREATEM signal. The Fugro company developed the time-domain airborne electromagnetic system, which installed compensation devices in the hardware to correct coil motion, and used notch filter with center frequency of 0.5Hz in the data processing. Buselli et al. (1998) proposed that a high-pass filter with a cut-off
frequency of 10Hz to filter out coil motion. Lemire et al. (2001) used spline interpolation and Lagrange optimization to remove low frequency noise. Yuan et al. (2013) introduced wavelet-based for signal baseline drift correction using sym8 wavelet with 10 layers. Because it is difficult to choose optimal wavelet basis function and the layer levels of wavelet decomposition, this method has poor adaptability. Fubo et al. (2017) focussed on EEMD method to distinguish and suppress motion-induced noise. But this method distorted reconstructed signals by deleting the components with motion-induced
noise directly.

N.E.Huang et al.(1998) proposed empirical mode decomposition (EMD) and Z. Wu et al.(2009) found EEMD. The EMD and EEMD method is a scale-adaptive time-domain method which is applied to non-linear, non-stationary signal





decomposition. For non-stationary signal processing, it is demanded to propose STFT and wavelet transform. But the requirement of signal characteristic above method is stationary in a specific window as same as the Fourier transform.

Different from previous methods, the major advantage of the EEMD is that the decomposition is derived from the signal itself. Therefore, the EEMD analysis is adaptive in contrast to the traditional methods where the decomposition functions are fixed in a specific window. Finally, the characteristics of the signal itself are not affected in the sifting process.

According to the characteristics of baseline wander for GREATEM signal, the EEMD adaptive filtering (EEMD-AF) is presented in this paper. This method consists of three steps.

step 1. The signal is decomposed into N-level IMF components and the residual component.

step 2. It is careful to use a low-pass filter for high-order IMF to filter off baseline wander signal.

step 3. The noise-free signal can be obtained by subtracting baseline wander from the noisy signal.

In the later section, compared with that of wavelet-based and EEMD with deleted higher-order components directly, the correctional result shows that the EEMD-AF method is a practical and effective method for removal of the baseline wander

on GREATEM signal.

## 2 Correction method of EEMD-AF

### 2.1 EMD methods

The EMD method decomposes the signal S(t) into N-level IMF components and a residual component. The EMD involves the adaptive decomposition of gave signal S(t) by means of a decomposition process called sifting algorithm. The term of

IMF is adopted because it represents the oscillation mode embedded in the data. The sifting process is defined by the following steps:

step 1. Identify levels of decomposition N, and $r_{j-1}(t)=S(t)$ as residual parameter;

step 2. Extract $IMF_j$;

(a) all extrema of $r_{j-1}(t)$;

(b) Interpolate local maxima and minima as the upper and lower envelopes separately by a cubic spline line. And compute "envelope" Emin(t) and Emax(t);

(c) Compute the average $m(t) = (Emin(t) + Emax(t))/2$;

(d) Extract the detail $D_i(t) = x(t) - m(t)$;

(e) Iterate step (a) to step (d) on the detail D(t) until satisfy stopping criterion, sd < ε. Once criterion is achieved. Detail D(t)

is considered as the effective $IMF_j(t)$, which can be considered as zero-mean generally. Calculate stopping criterion:

$$sd = \sum \frac{|D_{i-1}(t) - D_i(t)|^2}{D_{i-1}(t)^2} \tag{1}$$

step 3. Update residual: $r_j(t)=r_{j-1}(t) - IMF_j(t)$, the residual is deemed as the input for a new round of iterations;

step 4. Repeat step 2 with j until the value of j equal to N.

The stop criterion ε of the sifting process is set between 0.2 to 0.3. The result of the sifting procedure is that S(t) will be

decomposed into $IMF_j(t)$, j = 1, …N and residual $r_N(t)$.

$$S(t) = \sum_{j=1}^{N} IMF_j(t) + r_N(t) \tag{2}$$

## 2.2 EEMD methods

EEMD method is an improved method based on EMD algorithm to eliminate the mode mixing problem of EMD. Compared with the EMD method, the EEMD method resolves the mode mixing problem and achieves better performance by adding

white noise to the original signal (Z.Wu and N.E. Huang, 2004). The EEMD produces an ensemble of data sets by adding different realizations of a Gaussian distribution white noise with finite amplitude σ to the original data. The σ is standard deviation of white noise. The following, EMD method repeated NE times is applied to each data series of the ensemble to get $IMF_i(t)$. Finally, the $\widehat{IMF}_j$ is obtained by averaging the respective components in each realization to offset the impact of the Gaussian white noise.

$$\widehat{IMF}_j(t) = \frac{1}{NE} \sum_{i=1}^{NE} IMF_i(t) \tag{3}$$

where NE is the ensemble numbers. Finally, the result of the sifting procedure is that S(t) will be decomposed into $\widehat{IMF}_j(t)$, j = 1, …N and residual $r_N(t)$.

$$S(t) = \sum_{j=1}^{N} \widehat{IMF}_j(t) + r_N(t) \tag{4}$$

where σ is set between 0.05 to 0.2 and NE is set to 200. In this paper, we set σ and NE to 0.1 and 200 respectively.

## 110   2.3 EEMD-AF methods

The EEMD method is equivalent to a sifting filter, which sifts the signal S(t) from fast oscillations to slow oscillations for each IMF component. The lower-order IMF component mainly contains fast oscillations, meanwhile the higher-order IMF component mainly contains slow oscillations. The baseline wander is expected to be captured by higher-order IMFs of large indices. Simply removing the last several IMFs may introduce significant distortions of the reconstructed signal.

Thus, the baseline wander is distributed over the desired components in the last several IMFs. To remove the baseline wander, this method introduces a group of adaptive low-pass filter to process the last several IMFs successively. The sum of the output of these filters is regarded as the reconstructed baseline estimate. Finally, the noise-free signal can be obtained by subtracting an estimated baseline from the noisy signal.

We suppose the signal S(t) contained severe baseline wander. After processing the EEMD, S(t) will be decomposed into

IMFs which referred as to $a_k(t)$:

$$S(t) = \sum_{k=1}^{N} a_k(t) \tag{5}$$

where N is the number of IMFs. Then, it is important to find out which number of IMFs contributes to the baseline wander. Denote this number value as M. The $a_k(t)$ is processed from high-order to low-order by low-pass filter of $h_k(t)$. The output of this filter as:

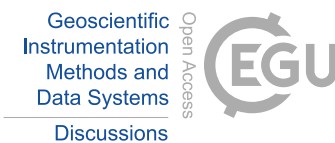

$b_k(t) = h_k(t) * a_k(t)$                                                                    (6)

where $*$ denotes the convolution. The $h_k(t)$ is the Butterworth low-pass filter, and the cut-off frequency is $\omega_k$. As the IMF order decreases, fewer slow oscillations components, but more signal components are contained in each IMFs. So, we design a group of adaptive low-pass filter with cut-off frequency decreases as IMF order decreases. The method processes the IMFs starting from the last, $a_N(t)$, filtered by first cut-off frequency $\omega_N(t)$. The cut-off frequency of the k-th filter is set as:

$\omega_{k-1} = \omega_k * \alpha$                                                            (7)

where $\alpha$ is set between 0.1 to 0.99 and $k = N, \cdots, 2, 1$. The filter output $b_k(t)$ contained low-frequency components are extracted from each IMF. Next, the output can be used to determine value of M. According to the characteristics of IMFs, the amplitude of the baseline should gradually decrease as the order decreases. As a result, to determine the value of M, we take the standard deviation $std(b_k)$ from $b_k(t)$ and evaluation coefficient $P_k$ regarded as stopping criterion.

$P_k = \dfrac{flip(std(b_k))}{\frac{1}{k-1}\sum_{i=1}^{k-1} flip(std(b_k))}$                    (8)

where $k = 1, 2, \cdots, N$, The operator *flip* is the flipped function. Set the threshold coefficient $\varepsilon$ and the value range from 0 to 0.1. If $P_k < \varepsilon$, the cut-off order $M = N + 1 - k$. In this process, we set $\omega_N$, $\alpha$, $\varepsilon$ to 10, 0.9 and 0.01, respectively. The sum of filter off IMF with orders from M+1 to N is regarded as the reconstructed baseline estimate.

$\widehat{b(t)} = \sum_{k=M+1}^{N} b_k(t)$                                                         (9)

Finally, to obtain reconstructed signal, baseline estimate is subtracted from the original signal.

$\widehat{S(t)} = S(t) - \widehat{b(t)}$                                                           (10)

### 3 Simulation data analysis

#### 3.1 Simulation data

In GREATEM system, the transmitter injects a bipolar square wave current into the ground, meanwhile the receiver and
front-mounted coil were installed on an aircraft to response to the vertical component of the induced electromotive force in a horizontal layered earth model (Nabighian et al, 1988). Responded signals are related to the size and depth of the underground conductor, the line length and current of the transmitter, equivalent area of the receiving coil, horizontal offset, flight altitude and so on. These parameters can be used to calculate the time domain response as clean signal in the horizontal layer earth model. In Fig. 1, the model parameters are as follows: the length of the transmitter line TX is 1000m on the
ground, transmitter current I is 10A with a frequency of 25Hz, equivalent area of the receiving coil RX is 1000m$^2$, offset location of receiver is 50m, flight altitude is 35m, the sample rate of receiver is 32kHz. In this paper, we consider three−layer earth model where parameters are shown in Table 1. In the end, we calculated vertical response decay curve and the corresponding time domain signal on a three-layer earth model.

Because of non-periodic and non-stationary characteristics of baseline wander, it is difficult to synthesize this noise from
simulation on the computer. The synthetic signal is obtained by superimposing the field baseline wander measured by the





inertial navigation system to clean signal system. Fig. 2(a) is synthetic noisy signal which obtained by adding baseline wander to clean signal with 10 seconds. Fig. 2(b) is the field baseline wander noise with 10 seconds.

### 3.2 Other methods of correction

#### 3.2.1 EEMD correction

In the EEMD method, the trend regarded as baseline estimate is expected to be captured by IMFs of large order. The reconstructed signal may therefore amount to the partial IMFs from lower-order to middle-order with ignored higher-order components directly (Patrick Flandrin et al., 2004).

#### 3.2.2 Wavelet-based correction

The wavelet-based method is based on multi-resolution decomposition analysis, which can employ commonly used wavelet
bases (e.g. Haar, db2, db4, db8, sym2, sym4, sym8) with 8 to 12 decomposition levels in processing data. After comparing result, the optimal wavelet basis is sym8, and the decomposition level is 10 (Yuan et al., 2013).

### 3.3 Performance of the correction and analysis

In this paper, in order to quantitatively assess the detrending quality between our method and other methods, we propose the signal-to-noise ratio (SNR) and mean square error (MSE) to evaluate the correction methods quantitatively in equation 11
and 12, in which $S(n)$ is synthetic clean signal, $\widehat{S(n)}$ is processed signal, L is the length of samples. The higher the SNR, the better the correction effect; the lower the MSE, the better the fitting result. There are comparison of the SNR and MSE of the synthetic noisy signal on before and after correction using EEMD-AF, wavelet-based and EEMD method, respectively.

$$SNR = 10\lg\left(\frac{\sum_{n=1}^{L} S^2(n)}{\sum_{n=1}^{L}(S(n)-\widehat{S(n)})^2}\right)$$                      (11)

$$MSE = \frac{\sum_{n=1}^{L}(S(n)-\widehat{S(n)})^2}{L}$$                       (12)

In Table 2, the correctional result of noisy signal with 60 seconds applied by three methods are shown, in which the term Noisy signal means SNR of synthetic noisy signal before correction. The SNR value shows that the three methods have a remarkable improvement in signal quality. It is proved that EEMD-AF and wavelet-based method had better correctional performance than EEMD. Quantitatively, EEMD-AF method yields SNR, which is significantly close to SNR achieved by the wavelet-based method. It can be shown that the EEMD-AF achieves correction performance as similar to the wavelet-
based method.

For further analysis, in the data processing of GREATEM system, response decay curve is related to the conductivity of underground geological bodies. Besides the above SNR comparison on time domain signal, the processed data above are used for stacking, averaging and extracting secondary field to build one test point on the whole survey path. Then the gates are approximately logarithmically spaced by 24 gates per point. Generally, the gates were referred to as channels.



To generate the anomaly curves profile image, we process the synthetic noisy data and correctional data using EEMD-AF and wavelet-based method. The anomaly curves profile generated from the clean signal responses are shown in Fig. 3(a), where represented paralleled line of 24 channels along with the test point. And the Fig. 3(b) is anomaly curves profile generated from the calculated noisy signal responses, where we can clearly identify the anomaly from Channel 14 to end due to the interference of baseline wander on the horizontal earth model. From Channel 20 to 24, they mixed each other. After

the processing using the wavelet-based method, the anomaly curves profile is shown in Fig. 3(c), where the anomaly is not accurately represented and the curves are similar to parallel each other. Fig. 3(d) is the anomaly curves profile using the EEMD-AF method, it is pretty obvious that the paralleled curves between the channels are better than the above method.

A typical comparison of SNR and MSE profile produced by the different method is illustrated along with the test point in Fig. 4, where SNR and MSE of the noisy signal is marked as reference (black solid line). In Fig. 4(a), the processing of noisy

signal shows that the stacking and averaging may produce the improved SNR. Quantitatively, the EEMD-AF and wavelet-based method yields SNR, which is significantly higher than value achieved by the EEMD method. It is observed that there are fluctuations of SNR using wavelet-based method (blue solid line) meanwhile there are stabilities of SNR using EEMD-AF method (red solid line). And in Fig. 4(b), the MSE curve indicates the same conclusion. Results from both methods also show that the EEMD-AF correctional method significantly outperforms the wavelet-based for non-stationary baseline

wander.

## 4. Field data analysis

In October 2018, an experimental GREATEM survey was performed to detect infiltration water in the refuse landfill of Longquanyi District, Chengdu in China. GREATEM system was developed by Chengdu University of Technology. The electrical source transmitter was fixed on the ground meanwhile the receiver system is mounted on the six rotor UAV. The

survey area and flight paths of the GREATEM system is shown in Fig. 5(b). The length of the transmitter line was 1100m on the ground, the transmitter waveform was bipolar square wave, transmitter current was 20A with 50% duty cycles at 5Hz. The receiver system made use of 24-bit Analog-to-Digital Converter and sample rate was 32kHz, and equivalent area of the receiving coil is 1000m$^2$. The transmitter line was set in the middle of flight paths and almost perpendicular to each other. The length of the flight line was 800m and spacing was 80m. The flight speed of the UAV was 2.5m/s, and the flight height

was 50m from the surface.

The amplitude of the response will decrease with the increase of transmitter–receiver offset. We choose measured data of flight path L4 for our processed, and the data of length of 60 seconds are shown in Fig. 6(a). We can significantly observe the baseline wander on the measured signal. And Fig. 6(b) is the correction result from the EEMD-AF method. By visual comparison of the data signals, the baseline wander is effectively eliminated by EEMD-AF.

Besides the above visual comparison, we produced anomaly curves profile image of the original measured data, the corrected data with the wavelet-based method and the EEMD-AF method, respectively. The gates are approximately


logarithmically spaced with 18 gates referred to as 18 channels per point. Fig. 7(a) is the anomaly curves profile generated from the measured raw data, Fig. 7(b) is the correctional data using the wavelet-based and Fig. 7(c) using the EEMD-AF correction. Based on the survey area are refuse landfill, we know that the geological structure of the flight area is layered earth, that there may be partial regions where the infiltration water was leaked.

Therefore, in Fig. 7(a), the higher responded anomaly curves reflected at 220m, 270m and 300m in the flight survey line, which will affect exploration elevation and the anomalies result on inversion because of the baseline wander existing in the original signal. It is obvious to observe the fake anomalies from Channel 10 to 15 and the interfere with each other from Channel 16 to 18. In Fig. 7(b) and (c), after using the baseline correction methods, the fake anomalies are reduced from Channel 10 to 15, and it is improved to interfere with each other from Channel 16 to 18. The Fig.7(c) shows that there is no interference between the channels while there is partial interference on Channel 16 to 18 especially in Fig.7(b). Comparison of Fig. 7(b) and (c), the results reveal that the performance of EEMD-AF method is significantly superior to the wavelet-based method to remove out baseline wander. In a word, the results confirm EEMD-AF method is an effective, practical correctional method.

## 5 Conclusion

Motion-induced noise was referred to as baseline wander that is an inevitable noise for GREATEM system with low frequency, large amplitude, non-periodic and non-stationary. The noise is caused by the receiving coil motion and always exists in the process of data acquisition to affect the measurement results severely, leading to inferior the exploration elevation and the fake anomalies result on inversion. Therefore, we proposed the improved EEMD method for baseline wander correction. The noisy signal is decomposed into N-level IMF components and residual component by EEMD method, and the baseline wander is distributed over the desired components in the last several IMFs, then a group of adaptive low-pass filter process last several IMFs successively. The sum of the filter output is reconstructed as a baseline estimate. Finally, the noise-free signal can be obtained by subtracting an estimated baseline wander from the noisy signal.

In this paper, through comparison of synthetic noisy data using correctional method of EEMD, wavelet-based and EEMD-AF, the SNR and MSE results show that the EEMD-AF method performance is significantly superior to the other methods. Furthermore, the same conclusion can be reached for the anomaly curves profile image. When processing field data, the baseline wander is effectively suppressed by EEMD-AF and wavelet-based methods. However, the results of comparison reveal that EEMD-AF method is significantly superior to the wavelet-based method to remove out baseline wander. Contrast with wavelet-based method, there is no interference between last three channels for data using EEMD-AF method on anomaly curves profile. And the decay curves of the whole survey line hold decay time 4.5ms more than wavelet-based method to improve the exploration elevation. These results also indicate that the improved EEMD method is a practical as well as effective method for removal of the baseline wander on GREATEM signal.



**Data availability**

In this paper, the data are not publicly accessible, because funder terms require to kept confidential for the original
geological data without cooperative licensing agreements.

**Author contribution**

Yuan L. and Song G. designed the method model and developed code. The author Saimin Z. designed the experiments and
carried it out with Hu H. and Pengfei X. . Yuan L. and Chunmei Y. performed the simulations and processed data. Yuan L.
prepared the manuscript with contributions from all co-authors.

**Competing interests**

There are no competing interests in this paper. And the authors declare that they have no conflict of interest.

**Acknowledgements**

This study was supported by Research on fixed wing time domain airborne electromagnetic measurement technology system
(2017YFC0601904). The authors thank the members of the project committee for their help.




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



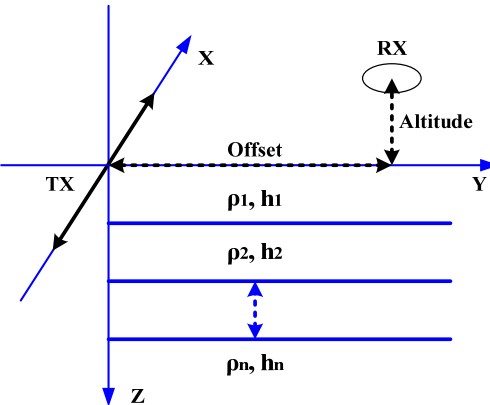

**Figure 1: GREATEM model based on three-layer earth model where parameters are shown in Table 1.**

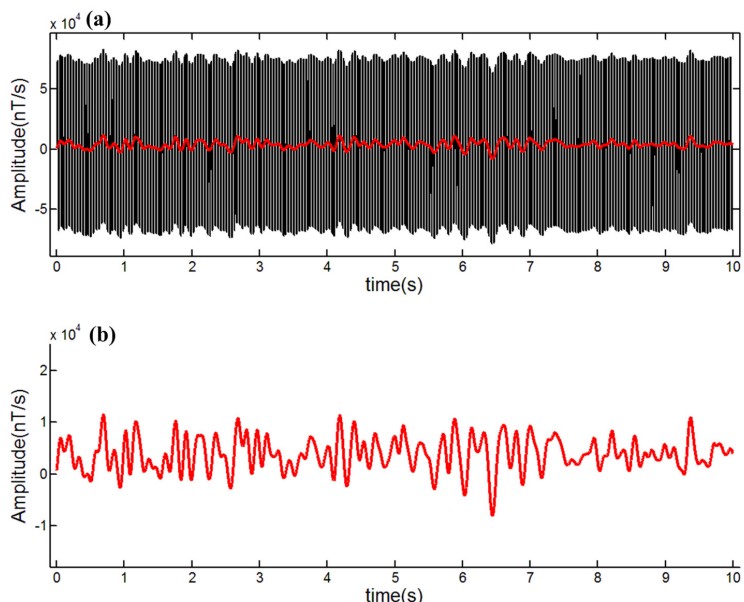

**Figure 2: The synthetic noisy signal and baseline wander. (a) The synthetic noisy signals; (b) field baseline wander.**



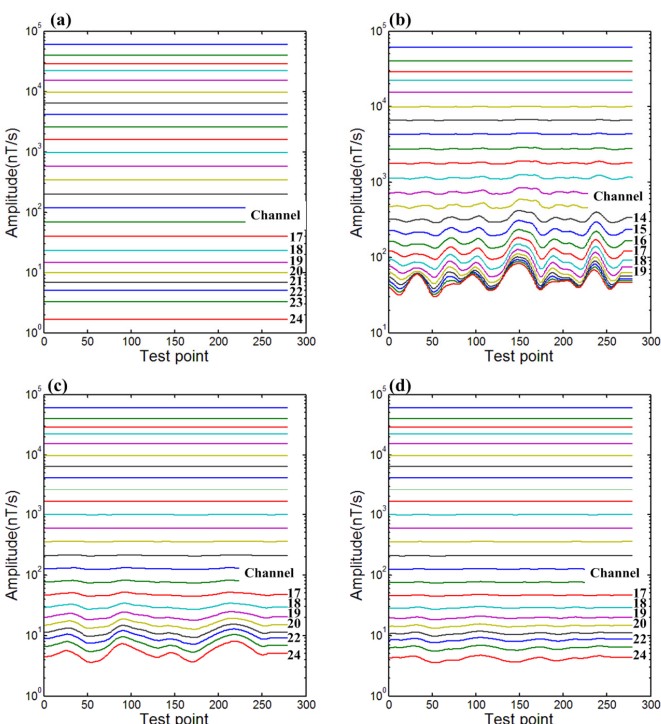

**Figure 3: Anomaly curves profile image generated from different datasets: (a) The clean signal from the theoretical model; (b)the noisy signal containing baseline wander; (c) the correctional data using wavelet-based method; (d) the correctional data using EEMD-AF method.**

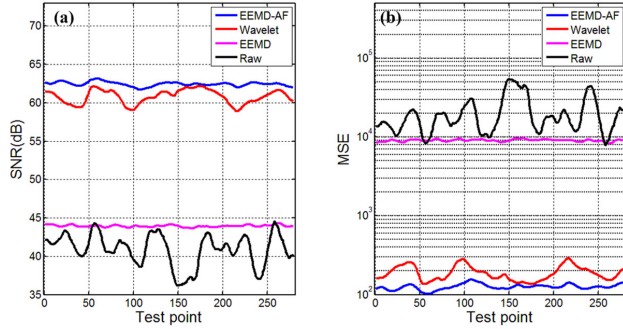


**Figure 4: Comparison of SNR and MSE profile produced by the different method. (a) SNR of different methods along with test point; (b) MSE of different methods along with the test point.**





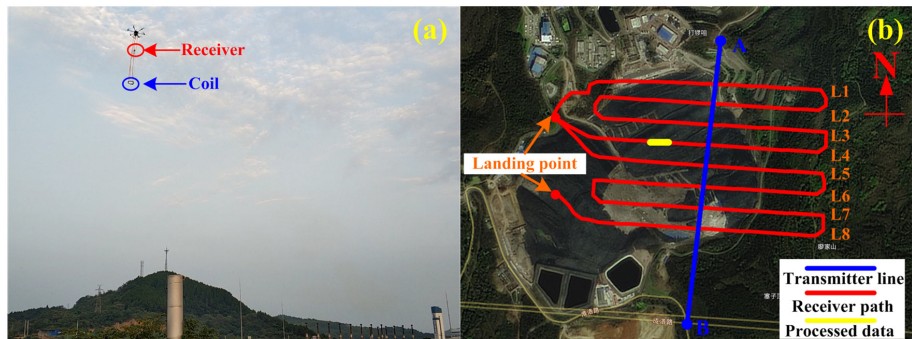

**Figure 5: The survey area and flight paths of the GREATEM system. (a) The receiver system is mounted on UAV along the paths.**
**(b) The blue line was the transmitter source and the red line was the survey path of the receiver. The flight heading was from east to west on the L4 path. The data of part of L4 (yellow solid line) was processed and the length of time was 60 seconds. (b) embedded the satellite images came from https://map.tianditu.gov.cn/ built by the National Geomatics Center of China.**

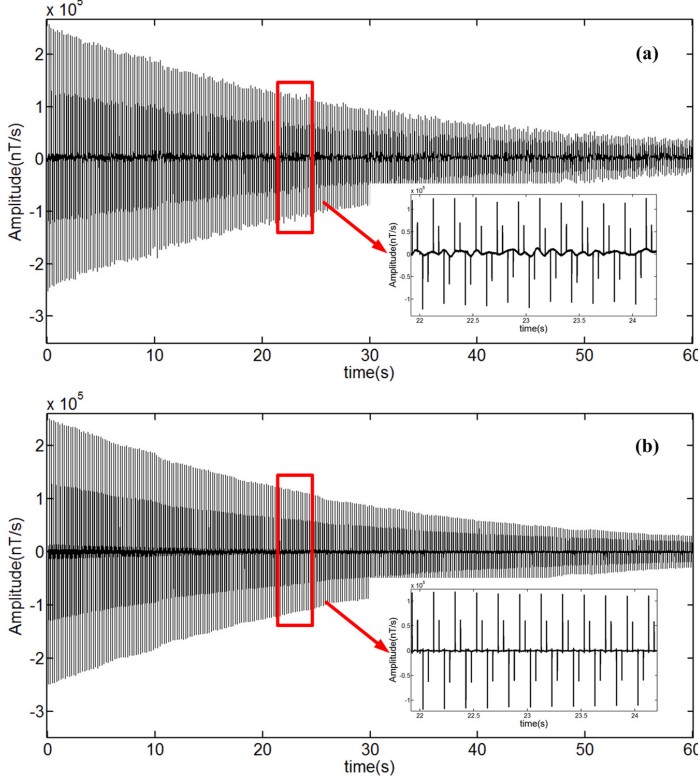



**Figure 6: Field data with baseline wander and correctional signal using the EEMD-AF method. (a) Field data acquisition by**
**GREATEM instruments; (b) corrected signal using the EEMD-AF method. The signal are magnified from 22 seconds to 24 seconds and shown at the lower right of each scheme.**

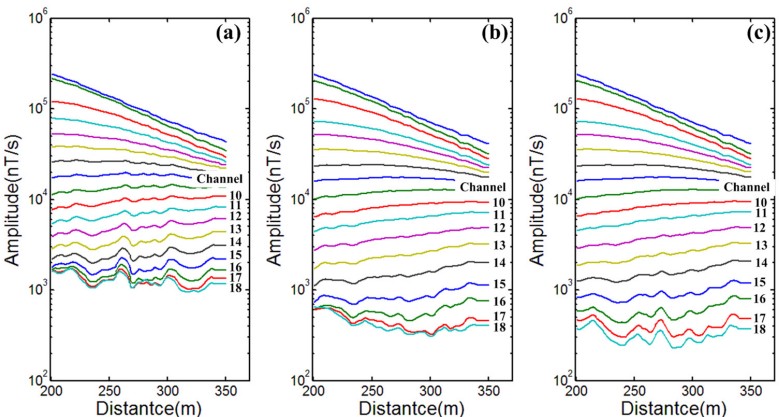

**Figure 7: Anomaly curves profile image generated from field data: (a) Profile of raw data.; (b) profile of data using wavelet-based method; (c) profile of data using EEMD-AF method.**

**Table 1: Parameters of the three-layer earth model**

| Parameter | Resistivity $\rho_n$ ($\Omega$ m) | Thickness $h_n$ (m) |
|-----------|-----------------------------------|---------------------|
| 1st | 150 | 100 |
| 2nd | 30 | 100 |
| 3th | 300 | |

**Table 2 Correctional result comparison of SNR of different method**

| Method | SNR(dB) |
|--------|---------|
| Noisy signal | 5.0810 |
| EEMD-AF | 48.1462 |
| EEMD | 35.1025 |
| Wavelet-based | 48.2513 |