# Peer review of "The baseline wander correction based on improved EEMD algorithm for grounded electrical source airborne transient electromagnetic signals"

_Geoscientific Instrumentation, Methods and Data Systems, 2020_

## Referee Comment (RC1) · Anonymous Referee #1 · 14 Jul 2020

The manuscript titled "The baseline wander correction based on improved EEMD algorithm for grounded electrical source airborne transient electromagnetic signals" by Yuan Li, Song Gao, Saimin Zhang, Hu He, Pengfei Xian, Chunmei Yuan, fit within the stated scope of the journal.

The inherent coil motion-induced noise in the GREATEM system always affect the measurement results. To improve the raw data, various methods are tried and a proposal has been made. The paper adds some knowledge in applying a method in the field of signal decomposition of electromagnetic signal into intrinsic components and filter high-order IMF component containing the base line wander.

[Figure]

The paper reads as if many new results are presented in the last section of the paper, and the reader is left interpreting these results as a summary rather than solid conclusion. I believe these results (including the figures) should be presented in a 'discussion' section separate from a 'conclusions' section. I believe much of the material is already in the paper, so it shouldn't be difficult for the authors to reorganize the latter sections of the paper accordingly.

The quality of the figures is good, but It should always be kept in mind that the description of the paper must be clear enough to favor the reader's understanding. To achieve this I would suggest also the following:

One should explain what each element that appears in the figures means, commenting on the meaning of axes and labels involved in the figures, which must be in accordance with the body of the manuscript text.

Finally, the references section is weak. It will be advisable to increase and update it somewhat.

---

## Referee Comment (RC2) · Anonymous Referee #2 · 6 Aug 2020

The manuscript discussed a new method for baseline wander correction of grounded electrical source airborne transient electromagnetic signals. Simulated and field results reveal that this new method is practical and effective for removal of the baseline wander. I believe that the conclusions of this manuscript have wider applications for non-periodic and non-stationary signal processing. However, I believe that the authors should address the following issues before being considered for publication in this journal. 1. In section 1, the authors state that the EEMD is different from STFT and wavelet transform. Could the authors provide more theoretical results to further strengthen the contrast of difference so as to prove that EEMD method is a scale-adaptive time-domain method that is applicable to non-stationary signal processing?

2. In Section 5, many results are presented and interpreted by the authors. However, no specific conclusions can be drawn from these results. I believe a better conclusion based on the available material in the manuscript should be provided in Section 5. Could the authors revise Section 5 and give some conclusions in the revision? 3. In Figures 5, could the authors zoom in and display the UAV and receiver instrument in Figure 5a and mark the direction of the flight path in Figure 5b? 4. Figures play the positive role in understanding the result of the method. However, professional terms marked in figures need further confirmation. The interpretation of the figure needs more details. 5. The language in this manuscript should be further improved. For example, "However, the signals are the superposition of useful signals and various noise signals." This needs to be further polished. 6. "In this paper, we propose improving method EEMD-AF based on ensemble empirical mode decomposition (EEMD)". What does AF mean? As always, abbreviations should always spelt out the first time you use them. Furthermore, a space should be kept between numbers and units such as 10Hz and 0.5Hz. 7. What is Section 3.2 for? It seems like a literature review that needs to be in the introduction? Section 3.2.1 and 3.2.2 should be combined?

---

## Author Comment (AC1) · 25 Aug 2020

Thank you very much for your comments. The comments will be very useful for this paper. Firstly, we will revise the 'conclusions' section and get solid conclusion. In this paper, some discussion will be added to the 'Field data analysis' section for anomaly curves profile image generated from different methods. Secondly, we have updated description of figure 1, 3, 5, 7 in accordance with manuscript. The interpretation of figure contains more details. Lastly, we will update and check references and formats.

[Figure]

[Figure]

Figure 1: GREATEM model based on three-layer earth model. The TX is length of the transmitter line on the ground and the line length is 1000 m, the current is 10 A, the frequency is 25 Hz. The RX is receiving coil and the equivalent area is 1000 m$^2$, the offset is 50 m, the flight altitude is 35 m, the sample rate of receiver is 32 kHz. The other model parameters are shown in Table 1.

**Fig. 1.** fig1

[Figure]

Figure 3: Anomaly curves profile image generated from different datasets. The simulation time of raw data is 60 s, and the stacking interval is 0.2 s therefore the number of the Test points is 300. In figure 3, (a) The clean signal from the theoretical model; (b) the noisy signal containing baseline wander; (c) the correctional data using wavelet-based method; (d) the correctional data using EEMD-AF method. The label 'Gate' marked in sub-figures represents the time gates from 1 to 24. Every time gate means different time width which increased logarithmically.

**Fig. 2.** fig3

[Figure]

Figure 5: The survey area and flight paths of the GREATEM system. (a) The receiver system is mounted on UAV along the paths; (b) The blue line was the transmitter source and the red line was the survey path of the receiver; (c) the receiver instruments; (d) the receiving coil with diameter of 50 cm. The flight heading was from east to west on the L4 path. The data of part of L4 (yellow solid line) was processed and the length of time was 60 seconds. The sub-figure (b) embedded the satellite images came from https://map.tianditu.gov.cn/ built by the National Geomatics Center of China.

**Fig. 3.** fig5

[Figure]

[Figure]

**Figure 7: Anomaly curves profile image generated from field data: (a) Profile of raw data.; (b) profile of data using wavelet-based method; (c) profile of data using EEMD-AF method. The length of time for raw data was 60 s and the flight speed of the UAV was 2.5 m/s therefore the offset distance was 150 m.**

**Fig. 4.** fig7

---

## Author Comment (AC2) · 26 Aug 2020

Thank you very much for your comments. The comments will be very useful for this paper. According to comments, we will revise some sections of the paper. 1.We have revised the contrast of difference on EEMD, STFT and wavelet transform methods. This part of the description is modified as: For non-stationary signal processing, it is necessary to propose the short time Fourier transform (STFT) and wavelet transform generally. The main method of STFT is to divide the signal into short time intervals where the signal is approximately stationary, and then perform Fourier transform with signal on each time interval to get the frequency distribution. The main method of the

wavelet transform is to utilize a variable-scale sliding window where the specific data is approximately stationary on signal. The width of window is variable for time and frequency domain. At the same time, because it is difficult to choose optimal wavelet basis function and the layer levels of wavelet decomposition by the signal itself, this method has poor adaptability. Therefore, the requirement of signal characteristic above method is stationary in a specific window as same as the Fourier transform. Different from previous methods, the major advantage of the EEMD is that the decomposition is derived from the signal itself. Therefore, the EEMD analysis is adaptive in contrast to the traditional methods where the decomposition functions are fixed in a specific window. Finally, the characteristics of the signal itself are not affected in the sifting process. 2. We will revise the 'conclusions' section and update references section. Some discussion will be added to the 'Field data analysis' section for anomaly curves profile image generated from different methods. 3. The receiver instruments will be showed on Fig 5 (c) and (d). Please refer to the figure behind comments. 4. We have updated description of figure 1, 3, 5, 7 in accordance with manuscript. The interpretation of figure contains more details. 5. The grammatical and term errors in the paper will be corrected carefully. 6. Section 3.2 describe the other method for subsequent analysis. Section 3.2.1 and 3.2.2 statement will be combined and added to specific referenced papers on Section Introduction.
* * *
Figure 1: GREATEM model based on three-layer earth model. The TX is length of the transmitter line on the ground and the line length is 1000 m, the current is 10 A, the frequency is 25 Hz. The RX is receiving coil and the equivalent area is 1000 m², the offset is 50 m, the flight altitude is 35 m, the sample rate of receiver is 32 kHz. The other model parameters are shown in Table 1.

**Fig. 1.** fig1

[Figure]

Figure 3: Anomaly curves profile image generated from different datasets. The simulation time of raw data is 60 s, and the stacking interval is 0.2 s therefore the number of the Test points is 300. In figure 3, (a) The clean signal from the theoretical model; (b) the noisy signal containing baseline wander; (c) the correctional data using wavelet-based method; (d) the correctional data using EEMD-AF method. The label 'Gate' marked in sub-figures represents the time gates from 1 to 24. Every time gate means different time width which increased logarithmically.

**Fig. 2.** fig3

[Figure]

Figure 5: The survey area and flight paths of the GREATEM system. (a) The receiver system is mounted on UAV along the paths; (b) The blue line was the transmitter source and the red line was the survey path of the receiver; (c) the receiver instruments; (d) the receiving coil with diameter of 50 cm. The flight heading was from east to west on the L4 path. The data of part of L4 (yellow solid line) was processed and the length of time was 60 seconds. The sub-figure (b) embedded the satellite images came from https://map.tianditu.gov.cn/ built by the National Geomatics Center of China.

**Fig. 3.** fig5

[Figure]

Figure 7: Anomaly curves profile image generated from field data: (a) Profile of raw data.; (b) profile of data using wavelet-based method; (c) profile of data using EEMD-AF method. The length of time for raw data was 60 s and the flight speed of the UAV was 2.5 m/s therefore the offset distance was 150 m.

**Fig. 4.** fig7

---

## Author Response (AR1)

[revised manuscript text omitted]

批注 [L2]: (1) Comments from Reviewer 1
(2) Author's response:
The authors update and check references.
(3) Author's changes:
The authors update and check references, and add two references from Section 3.2 by the removal of section 3.2.

N.E.Huang et al.(1998) proposed empirical mode decomposition (EMD) and Z. Wu et al.(2009) raised EEMD. The EMD and EEMD method is a scale-adaptive time-domain method which is applied to non-linear and non-stationary signal decomposition. For non-stationary signal processing, it is necessary to propose short time Fourier transform (STFT) and wavelet transform generally. The main method of STFT is to divide the signal into short time intervals where the signal is approximately stationary, and then perform the Fourier transform of signal on each time interval to get the frequency distribution. And the main method of the wavelet transform is to utilize a variable-scale sliding window where the specific data is approximately stationary on signal. The width of window is variable for time and frequency domain. However, because it is difficult to choose optimal wavelet basis function and the layer levels of wavelet decomposition by the signal itself, this method has poor adaptability. Therefore,  the requirement of signal characteristic above method is stationary in a specific window as same as the Fourier transform.

批注 [L3]: (1) Comments from Reviewer 2
(2) Author's response:
The authors add section on the contrast of STFT and wavelet transform methods.
(3) Author's changes:
The contrast of STFT and wavelet transform method required signal is stationary in a specific window as same as the Fourier transform. The EEMD is that the decomposition is derived from the signal itself in next section.

Different from previous methods, the major advantage of the EEMD is that the decomposition is derived from the signal itself. Therefore, the EEMD analysis is adaptive decomposition in contrast to the traditional methods  that the decomposition functions are fixed in a specific window throughout the processing. In addition, the characteristics of the signal itself are not affected in the sifting process.

According to the characteristics of baseline wander for GREATEM signal, the EEMD  method consists of three steps.

step 1. The signal is decomposed into the N-level IMF components and the residual component by the EEMD method.

95 step 2. It is careful to use an adaptivea low-pass filter for higher indexhigh-order IMFs to get filter off baseline wander estimatesignal.

step 3. The de-noisednoise-free signal can be obtained by subtracting baseline wander from the noisy signal.

In the later section, compared with that of wavelet-based and EEMD without the higher indexwith deleted higher-order components directly, the correctional result shows that the EEMD-AF method is a practical and effective method for
100 suppressionremoval of the baseline wander on GREATEM signal.

**2 Correction method of EEMD-AF**

**2.1 EMD methods**

The EMD method decomposes the signal $S(t)S(t)$ is decomposed into N-level IMF components and a residual component by the EMD method. The EMD involves the adaptive decomposition of gave signal $S(t)S(t)$ by means of a decomposition
105 process called the sifting processalgorithm. The term of IMF is adopted because it represents the oscillation mode embedded in the data. The sifting process is defined by the following steps:

step 1. Identify levels of decomposition N, and $r_{j-1}(t) = S(t)$ $r_{j-1}(t)=S(t)$ as residual parameter;

step 2. Extract $IMF_j$$IMF_j$;

(a) all extrema of $r_{j-1}(t)$$r_{j-1}(t)$;
110 (b) Interpolate local maxima and minima as the upper and lower envelopes separately by a cubic spline line. And compute "envelope" $E_{min}(t)$Emin(t) and $E_{max}(t)$Emax(t);

(c) Compute the average component $m(t) = (E_{min}(t) + E_{max}(t))/2$m(t) = (Emin(t) + Emax(t))/2;

(d) Extract the detail component $D_i(t) = x(t) - m(t)$;

(e) Iterate step (a) to step (d) on the detail component $D(t)$D(t) until the stopping criterion satisfy thresholdsatisfy stopping

[revised manuscript text omitted]

批注 [L4]: (1) Comments from Reviewer 2
(2) Author's response:
Section 3.2 describe the other method for subsequent analysis.
Section 3.2.1 and 3.2.2 statement will be combined and added to specific referenced papers on Section Introduction.
(3) Author's changes:
The content of section 3.2 extracted and changed to Section Introduction as reference.

[revised manuscript text omitted]
. AndFurthermore, the same conclusion can be reached for from the anomaly curves profile image. Furthermore,When processing in field data processed, the baseline wander is effectively suppressed by EEMD-AF and wavelet-based methods. Because there is no interference between the last few Gates, However, the results of comparison of the anomaly curves profile image reveals that EEMD-AF method is significantly better thansuperior to the wavelet-based method to remove out baseline wander. Contrast with wavelet-based method, there is no interference between last three channels for data using EEMD-AF method on anomaly curves profile. And the decay curves of the whole survey line hold decay time 4.5ms more than wavelet-based method to improve the exploration elevation. These results also indicate that the EEMD-AFimproved EEMD method is a practical as well as effective method for the suppressionremoval of the baseline wander on GREATEM signal.

**Data availability**

In this paper, the data are not publicly accessible, because funder terms require to kept confidential for the original geological data without cooperative licensing agreements.

批注 [L5]: (1) Comments from Reviewer 1 and 2
(2) Author's response:
The authors revise the 'conclusions' section and get solid conclusion, some discussion will be added to the 'Field data analysis'.
(3) Author's changes:
The 'conclusions' section is revised and some discussion will be added to the 'Field data analysis'.

**Author contribution**

First, Yuan L. and Song G. designed the method model and developed code. Second,  the author Saimin Z. designed the field experiments and carried it out with Hu H. and Pengfei X. . Third, Yuan L. and Chunmei Y. performed the simulations and processed data. Finally, Yuan L. prepared the manuscript with contributions from all co-authors.

315 **Competing interests**

There are no competing interests in this paper. And the authors declare that they have no conflict of interest.

**Acknowledgements**

This study was carried out within the project ' Research on fixed wing time domain airborne electromagnetic measurement technology system (2017YFC0601904)' supported by the Institute of Geophysical and Geochemical Exploration, Chinese Academy of Geological Sciences. The authors thank the members of the project committee for their help.

The description of figure contains more details for interpretation.

[Figure]

[Figure]

360     **Figure 2: The** synthetic **noisy signal and baseline wander** signal whose duration is 10 s**. (a) The** synthetic **noisy signals; (b)** the **field baseline wander** measured by the inertial navigation system**.**

[Figure]

[Figure]

**Figure 3: Anomaly curves profile image generated from different  datasets. The duration of raw data is 60 s and the stacking interval is 0.2 s therefore the number of Test points is 300. (a) The clean signal from the theoretical model; (b) the noisy signal containing baseline wander; (c) the correctional signal using wavelet-based method; (d) the correctional signal using EEMD-AF method. The label Gate marked in each figure represents the number of time gates from 1 to 24. Every specific number of time gate means different time width which increased logarithmically.**

365

批注 [L7]: (1) Comments from Reviewer 1 and 2
(2) Author's response:
The authors updated description of figure 1, 3, 5, 7 in accordance with manuscript. The interpretation of figure contains more details.
(3) Author's changes:
The description of figure contains more details for interpretation.

[Figure]

**Figure 4: Comparison of SNR and MSE profile produced by  the different method. (a)  SNR  different ; (b)  MSE **

[Figure]

[Figure]

375 **Figure 5: The survey area and flight paths** . **(a) The receiver system is mounted on UAV along**  **the paths** ; **(b)**  **the blue line was the transmitter source and the red**  **curves were**  **the survey path**s **of the receiver.**, **the lines of L1 through L8 represent different paths and the orange dots represent the landing point for UAV; (c) the receiver instruments; (d) the receiving coil with diameter of 50 cm. The flight heading was from east to west on the L4 path. The data of part of L4 (yellow**  **solid line)**  **will be processed and the** duration 380  **was 60 s**. **The satellite images**  **came from https://map.tianditu.gov.cn/ built by the National Geomatics Center of China.**

批注 [L8]: (1) Comments from Reviewer 1 and 2
(2) Author's response:
The authors updated description of figure 1, 3, 5, 7 in accordance with manuscript. The interpretation of figure contains more details.
(3) Author's changes:
The description of figure contains more details for interpretation.

批注 [L9]: (1) Comments from Reviewer 2
(2) Author's response:
The receiver instruments will be showed on Fig 5 (c) and (d).
(3) Author's changes:
The receiver instruments will be showed on Fig 5 (c) and (d). And the description of figure contains more details for interpretation.

[Figure]

**Figure 6:** field data  60 s   baseline wander and correctional   the EEMD-AF method. (a)   data   by   instruments; **(b)**   data  using the EEMD-AF method.   is magnified  and shown at the lower right of each  .

385

[Figure]

**Figure 7: Anomaly curves profile image generated from field data on different methods.** (a) The  profile of raw data; (b) the profile of data using wavelet-based method; (c) the profile of data using EEMD-AF method. **Because the duration of raw data was 60 s and the flight speed of the UAV was 2.5 m/s, the offset distance was 150 m.**

批注 [L10]: (1) Comments from Reviewer 1 and 2
(2) Author's response:
The authors updated description of figure 1, 3, 5, 7 in accordance with manuscript. The interpretation of figure contains more details.
(3) Author's changes:
The description of figure contains more details for interpretation.

**Table 1: Parameters of the three-layer earth model**

| Parameter | Resistivity $\rho_n$ ($\Omega$ m) | Thickness $h_n$ (m) |
|---|---|---|
| 1st | 150 | 100 |
| 2nd | 30 | 100 |
| 3th | 300 | |

**Table 2 Correctional result comparison of SNR of different methods**

| Method | SNR(dB) |
|---|---|
| Noisy signal | 5.0810 |
| EEMD-AF | 48.1462 |
| EEMD | 35.1025 |
| Wavelet-based | 48.2513 |